

# Spreading of entanglement and correlations after a quench with intertwined quasiparticles

**Alvise Bastianello[1] and Pasquale Calabrese[1,2]**

**1** SISSA & INFN, via Bonomea 265, 34136 Trieste, Italy
**2** The Abdus Salam International Centre for Theoretical Physics,
Strada Costiera 11, 34151 Trieste, Italy

## Abstract

We extend the semiclassical picture for the spreading of entanglement and correlations to quantum quenches with several species of quasiparticles that have non-trivial pair correlations in momentum space. These pair correlations are, for example, relevant in inhomogeneous lattice models with a periodically-modulated Hamiltonian parameter. We provide explicit predictions for the spreading of the entanglement entropy in the space-time scaling limit. We also predict the time evolution of one- and two-point functions of the order parameter for quenches within the ordered phase. We test all our predictions against exact numerical results for quenches in the Ising chain with a modulated transverse field and we find perfect agreement.

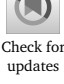
# 1   Introduction

During the last decade, the study of the non-equilibrium dynamics after a quantum quench (i.e. after an abrupt change of a parameter in a quantum Hamiltonian) has been the subject of intense theoretical and experimental investigations, see e.g. Refs. [1–4] as reviews on the subject. One of the main issues concerned the nature of the stationary state that describes local properties of the system. Nowadays we have a rather clear understanding of this stationary state: a generic system for long time attains a thermal state [5–11] while an integrable model relaxes to a generalised Gibbs ensemble [12–20] (also many-body localised systems have very peculiar non-equilibrium features [21–23]).

Conversely, the approach to the stationary state and the exact time evolution of physical observables remain less generally understood problems, in spite of a very intense activity. While some approximate numerical and analytical methods to tackle the problem in interacting integrable models exist (see, e.g., Refs. [24–26]), exact analytical results are scarse even for free systems: only few first principle calculations have been worked out up to a final analytic form [27–37]. In this respect, the quasiparticle picture [38,39] proved to be an extremely valuable tool. Although it is not an ab-initio technique, it provides a qualitative and quantitive understanding of the time evolution of some observables under specific conditions. It has been introduced to explain the entanglement evolution in the scaling limit after a quantum quench [38] and originally tested against the exact results in conformal field theories [38–41], in free models [29,38,39,42–51], and against many numerical simulations [52–59]. Only very recently these concepts have been used to quantitatively predict the time evolution of the entanglement entropy in generic interacting integrable systems [60–62]. In the field theoretical context, it has also been shown that the quasiparticle picture can be used to understand the time evolution of the one- and two-point functions of the order parameter [63, 64] (more generically of correlations of primary operators whose expectation value is non-vanishing in the initial states [63,64]). Some results in the very few analytically treatable free models are compatible with this picture [30,31], but the general regime of applicability of these ideas to generic operators is not clear [65–67], even within the realm of quadratic models.

A central object of this paper is the entanglement entropy [68–70]

$$S_A \equiv -\mathrm{Tr}\rho_A \ln \rho_A \,, \tag{1}$$

where $\rho_A \equiv \mathrm{Tr}_{\bar{A}}|\psi\rangle\langle\psi|$ is the reduced density matrix of a subsystem $A$ (having $\bar{A}$ as complement) of a system in a pure state $|\psi\rangle$. Its time evolution plays a crucial role in the understanding of the non-equilibrium dynamics of isolated quantum systems. Indeed, the growth of the entanglement entropy in time has been related to the efficiency of tensor network algorithms [71–76] such as the time dependent density matrix renormalisation group. Furthermore, the extensive value (in subsystem size) reached by the entanglement entropy at long time has been understood as the thermodynamic entropy of the ensemble describing stationary local properties of the system [60–62, 77–84].

Within the quasiparticle description, the initial state is regarded as a source of entangled quasiparticles which ballistically propagate across the system and carry entanglement. The structure of the pre-quench state in terms of the post-quench excitations is essential in dragging quantitative predictions for the spreading of entanglement. For example, in Ref. [29] the XY spin-chain has been considered

$$H_{XY} = -\sum_{j=1}^{N}\left[\frac{1+\gamma}{4}\sigma_j^x\sigma_{j+1}^x + \frac{1-\gamma}{4}\sigma_j^y\sigma_{j+1}^y + \frac{h}{2}\sigma_j^z\right], \tag{2}$$

and quenched in the magnetic field $h^{t<0} \to h$, starting from the ground state at $h^{t<0}$. The XY model is diagonalised in terms of spinless fermions through a Jordan-Wigner transformation:

in the thermodynamic limit $N \to \infty$ the momentum is continuous, thus the fermions obey $\{\eta(k), \eta^\dagger(q)\} = \delta(k-q)$, and

$$H_{XY} = \int_{-\pi}^{\pi} dk\, E(k)\eta^\dagger(k)\eta(k) + \text{const.} \tag{3}$$

The prequench ground state, identified with the vacuum of the prequench modes $|0_{h^{t<0}}\rangle$, is readily written in terms of the post quench vacuum $|0_h\rangle$ in the form of a squeezed state

$$|0_{h^{t<0}}\rangle \propto \exp\left[ -\int_0^\pi dk\, \mathcal{K}(k)\eta^\dagger(k)\eta^\dagger(-k) \right]|0_h\rangle\,, \tag{4}$$

with $\mathcal{K}$ a non trivial function of $h$ and $h^{t<0}$, its specific form being irrelevant for our purposes. Squeezed states are common in quenches in free theories, due to the fact that the pre and post quench modes are usually connected through a Bogoliubov rotation. The form of Eq. (4) is rather appealing: the quench modes are excited in pairs of opposite momenta, distinct pairs being created independently. In this case, the entanglement growth is well described by the following quasiparticle picture [38].

The initial state is regarded as a source of quasiparticles, homogeneously distributed in space. After the quench, each particle ballistically propagates with velocity $v(k) = \partial_k E(k)$. Pairs originating at different positions or with different momentum are unentangled, only particles belonging to the same pair of momentum $(k, -k)$ and originating in the same position are entangled. Given the partition of the system $A \cup \bar{A}$ and considering a time $t$, only pairs such that one quasiparticle belongs to $A$ and the other to $\bar{A}$ contribute to the entanglement, their contribution being additive. In the case where $A$ is chosen to be an interval of length $\ell$, the entanglement entropy, in the space-time scaling limit $t, \ell \to \infty$ with $t/\ell$ fixed, has the following scaling form

$$S_A(t) = 2t \int_{2|v(k)|t<\ell} \frac{dk}{2\pi} |v(k)|s(k) + \ell \int_{2|v(k)|t\geq\ell} \frac{dk}{2\pi} s(k)\,, \tag{5}$$

where $s(k)$ is the contribution to the entanglement associated with each pair. In the standard homogeneous situation, the weight $s(k)$ can be fixed by requiring that for $t \to \infty$ the entanglement entropy density matches the thermodynamic one of the post quench steady state [60–62]

$$s(k) = -n(k)\log n(k) - \left[1 - n(k)\right]\log\left[1 - n(k)\right]\,, \tag{6}$$

being $n(k) = |\mathcal{K}(k)|^2/(1 + |\mathcal{K}(k)|^2)$ the density of excitations of momentum $k$, i.e. $\langle\eta^\dagger(k)\eta(q)\rangle = \delta(k-q)n(k)$. The generalisation to several, *uncorrelated*, particle species is obvious, one has just to sum over the contributions of each independent species. This is a simple further step which allows us to describe quenches in truly interacting models where several particle species may be present. Indeed, in Ref. [60] quenches in the XXZ spin-chain have been considered, and the quasiparticle picture (5) (through a suitable dressing of the velocity $v(k)$ and the weight $s(k)$) has been found to be correct. Interestingly, also Eq. (6) has a very simple semiclassical interpretation. Indeed, focussing on a given momentum $k$, $s(k)$ is just the entropy (i.e. the logarithm of the number of equivalent microstates) of a mode which is occupied with probability $n(k)$ and empty with probability $1 - n(k)$.

The main physical feature for the entanglement evolution captured by Eq. (5) is the so-called light-cone spreading, i.e. a linear increase at short time followed by saturation to an extensive value in $\ell$. Indeed, if a maximum velocity $v_{\max}$ for the quasiparticles exists, then since $|v(k)| \leq v_{\max}$, the second term vanishes when $2v_{\max}t < \ell$, and the first integral is over all positive momenta, so that $S_A(t)$ is strictly proportional to $t$. On the other hand as $t \to \infty$,

the first term is negligible and $S_A(\infty)$ is proportional to $\ell$. Actually, analytical, numerical and experimental results [54, 80–82, 84–90] suggest that such a light-cone spreading is more generically valid that what suggested by the quasiparticle picture.

At this point, it must be clear that the physical assumptions behind validity of Eq. (5) for the entanglement evolution is that quasiparticles must be produced *uniformly in space and in uncorrelated pairs of opposite momenta*. Given the large success of Eq. (5) in describing the entanglement evolution in the scaling regime, a lot of recent activity has been devoted to understand how Eq. (5) gets modified when some of these assumptions are weakened. For example, the quasiparticle picture has been extended to large-scale inhomogeneous setups [51] in which the initial state is regarded as a *non uniform* source of quasiparticles which ballistically propagate for $t > 0$. While the final expression (5) is slightly modified in order to keep in account the initial inhomogeneity, the core of the result is still Eq. (6), where $n(k)$ is promoted to have a weak spatial dependence and the semiclassical interpretation of the entanglement weight still holds true.

Another setup in which the semiclassical interpretation can be applied, but which lays outside the usual framework of uncorrelated pairs, has been studied in Ref. [48] in a free-fermion model. In that case, the homogeneous initial state was populated with excitations of $n$ different species, with a constraint on the sum of the excitation densities $\sum_{i=1}^{n} n_i(k) = 1$ which ultimately introduces non trivial correlations. However, this constraint is classical in nature and it does not spoil the semiclassical interpretation of the entanglement entropy. As a matter of facts, the constraint changes the form of Eq. (6), which nevertheless can still be viewed as the entropy of fermions obeying the extra condition. In particular, the entanglement growth is fully determined in terms of the excitation densities $\{n_i(k)\}_{i=1}^{n}$ and of the velocities of each species $\{v_i(k)\}_{i=1}^{n}$ with no other information required.

In this work we investigate those situations where the initial state is populated by several quasiparticle species, which are non trivially *quantum-correlated*. The presence of true quantum correlation among the excitations forces us to dismiss the simple entanglement weight (6) together with its classical interpretation. However, despite this lack of classicality, the quasiparticle paradigm will still hold true and the entanglement growth (in the scaling limit) is fully determined in terms of ballistically-propagating localised excitations.

In particular, we are interested in free Hamiltonians possessing $n$ species of excitations (assumed to be fermionic for concreteness)

$$H = \sum_{i=1}^{n} \int_{-B}^{B} \mathrm{d}k \, E_i(k) \eta_i^\dagger(k) \eta_i(k). \tag{7}$$

Each mode has its own group velocity $v_i(k) = \partial_k E_i(k)$ and we assume the existence of a single Brillouin zone $[-B, B]$. The initial state $|\Psi\rangle$ is taken to be a non trivial generalisation of the single species squeezed state (4), i.e.

$$|\Psi\rangle \propto \exp\left[ -\int_0^B \mathrm{d}k \sum_{i,j=1}^{n} \mathcal{M}_{i,j}(k) \, \eta_i^\dagger(k) \eta_j^\dagger(-k) \right] |0\rangle, \tag{8}$$

where $|0\rangle$ is the vacuum $\eta_i(k)|0\rangle = 0$. This class of states is gaussian, i.e. the knowledge of all the correlation functions can be reduced, by mean of a repeated use of the Wick Theorem, to the two-point correlators

$$\langle \eta_i^\dagger(k) \eta_j(q) \rangle = \delta(k-q) \mathcal{C}_{i,j}^{(1)}(k), \qquad \langle \eta_i^\dagger(k) \eta_j^\dagger(q) \rangle = \delta(k+q) \mathcal{C}_{i,j}^{(2)}(k), \tag{9}$$

where the correlation matrices $\mathcal{C}^{(1)}(k)$, $\mathcal{C}^{(2)}(k)$ have dimension $n \times n$ and are functions of $\mathcal{M}(k)$, the specific relation being inessential for our purposes. The contact point with the

previous literature can be made in the case where $\mathcal{C}^{(1)}(k)$ and $\mathcal{C}^{(2)}(k)$ are diagonal on the particle species. States in the form (8) are not rare, making our generalisation more than a mere academic question. For example, we can revert to suitable inhomogeneous quenches in free models in order to realise states such as (8), as we explain thereafter.

Our quasiparticle ansatz will be formulated in full generality without looking at a precise model, however we ultimately rely on the periodically-modulated inhomogeneous Ising chain as a convenient benchmark

$$H = -\frac{1}{2} \sum_{j=1}^{N} \left[ \sigma_j^x \sigma_{j+1}^x + h_j \sigma_j^z \right], \tag{10}$$

where we require $h_j$ to be periodic with period $n$

$$h_j = h_{j+n}. \tag{11}$$

The periodicity of the magnetic field effectively splits the original lattice into $n$ sublattices coupled in a non trivial way, each one with lattice spacing $n$ so that the Hamiltonian (10) can be then diagonalised in terms of $n$ particle species $\{\eta_i(k)\}_{i=1}^n$. Starting in the ground state for a given set of magnetic fields and quenching towards different $\{h_j\}_{j=1}^n$ creates initial states in the form (8). Of course, in the scaling region where the quasiparticle description holds true, the lengthscale of the inhomogeneity is negligible: such a quench can be regarded as being homogeneous with several species of quasiparticles.

This work is organised as it follows. in Section 2 we present a general discussion of our quasiparticle ansatz for states in the form (8). In Section 3 we benchmark our predictions in the inhomogeneous Ising model, providing the details of its solution. In Section 4 we provide a quasiparticle description for the time evolution of the correlators of the order parameter in the inhomogeneous Ising model, thus generalising the results of Ref. [30, 31]. In Section 5 we gather our conclusions. Two appendices support the main text with some technical details.

## 2 The quasiparticle prediction for the entanglement entropy spreading

In this section we present our result for the time evolution of the entanglement entropy. Of course, being a quasiparticle prediction, its derivation is not rigorous, but relies on reasonable physical arguments and on the experience gained from the existing literature. Our arguments are somehow related to those of Ref. [51], where the standard quasiparticle picture for a single species and pair excitations has been extended to weakly inhomogeneous non-equilibrium protocols. For definiteness, we focus on a a bipartition $A \cup \bar{A}$ where $A$ is an interval of length $\ell$. Following the standard quasiparticle picture, we regard the initial state as a source of excitations such that *i)* quasiparticles generated at different spatial points are not entangled, *ii)* quasiparticles associated with pairs of different momentum are not entangled. As usual [38, 51], we further assume the contribution to the entanglement of unentangled pairs to be additive

$$S_A(t) = \int_{-\infty}^{\infty} \mathrm{d}x \int_0^B \frac{\mathrm{d}k}{2\pi} s_A(x, k, t). \tag{12}$$

Here, $s_A(x, k, t)$ is the contribution to the entanglement at time $t$ given by the quasiparticles originated at $t = 0$ in position $x$ and with momentum $\pm k$ (the momentum integration in (12) runs on positive values in order to avoid double counting). The difficulty, as well as the main result of our investigation, is finding the correct ansatz for $s_A(x, k, t)$: we propose to construct

$s_A(x, k, t)$ out of suitable finite-dimensional ancillary Hilbert spaces and partitions thereof. To each position $x$ and momentum $k$ we associate a Hilbert space constructed as a Fock space starting from a vacuum $|0^{x,k}\rangle$, acting with $2n$ fermions $\{f_j^{x,k}, [f_i^{x,k}]^\dagger\} = \delta_{i,j}$; such a Hilbert space has dimension $2^{2n}$. We recall that $n$ is the number of different species of quasiparticles. We have in mind the following, suggestive, correspondence

$$f_i^{x,k} \longleftrightarrow \eta_i(k), \qquad f_{i+n}^{x,k} \longleftrightarrow \eta_i(-k). \tag{13}$$

In order to make clearer and precise such a statement, we consider a state in the ancillary Hilbert space encoded in a density matrix $\rho^{x,k}$ such that: *i)* it is gaussian in the fermions $f_i^{x,k}$ (i.e. the Wick Theorem holds true); *ii)* its correlators are the same of the corresponding modes. More specifically, let us organise the $\eta_i(k)$ modes and the fermions $f_i^{x,k}$ in single vectors as

$$\Gamma(k) = \begin{pmatrix} \eta_1(k) \\ \dots \\ \eta_n(k) \\ \eta_1(-k) \\ \dots \\ \eta_n(-k) \\ \eta_1^\dagger(k) \\ \dots \\ \eta_n^\dagger(k) \\ \eta_1^\dagger(-k) \\ \dots \\ \eta_n^\dagger(-k) \end{pmatrix}, \qquad \mathcal{F}_{x,k} = \begin{pmatrix} f_1^{x,k} \\ \dots \\ f_n^{x,k} \\ [f_1^{x,k}]^\dagger \\ \dots \\ [f_n^{x,k}]^\dagger \end{pmatrix}. \tag{14}$$

We then consider the correlation functions $\langle \Gamma(k)\Gamma^\dagger(q)\rangle$ and $\langle \mathcal{F}_{x,k}\mathcal{F}_{x,k}^\dagger\rangle_{\rho^{x,k}}$, where the first expectation value is taken with respect to the state in Eq. (8), while the second on the ancillary Hilbert space on the density matrix $\rho^{x,k}$ which is defined in such a way to satisfy (we recall that momenta are positive)

$$\langle \Gamma(k)\Gamma^\dagger(q)\rangle = \delta(k-q)\mathcal{C}(k), \qquad \mathcal{C}(k) = \langle \mathcal{F}_{x,k}\mathcal{F}_{x,k}^\dagger\rangle_{\rho^{x,k}}. \tag{15}$$

We finally impose $\langle f_i^{x,k}\rangle_{\rho^{x,k}} = 0$. Given that the density matrix $\rho^{x,k}$ is Gaussian, having established its one- and two-point functions completely fixes the density matrix itself. The matrix $\mathcal{C}(k)$ has dimension $4n \times 4n$ and may be written in terms of the correlation matrices $\mathcal{C}^{(1)}(k)$ and $\mathcal{C}^{(2)}(k)$ in (9) as

$$\mathcal{C}(k) = \left( \begin{array}{c|c|c|c} \mathrm{Id} - \mathcal{C}^{(1)}(k) & 0 & 0 & [\mathcal{C}^{(2)}(-k)]^\dagger \\ \hline 0 & \mathrm{Id} - \mathcal{C}^{(1)}(-k) & [\mathcal{C}^{(2)}(k)]^\dagger & 0 \\ \hline 0 & \mathcal{C}^{(2)}(k) & \mathcal{C}^{(1)}(k) & 0 \\ \hline \mathcal{C}^{(2)}(-k) & 0 & 0 & \mathcal{C}^{(1)}(-k) \end{array} \right). \tag{16}$$

In particular, thanks to the block structure

$$\mathcal{C}(k) = \left( \begin{array}{c|c} \mathrm{Id} - M & N \\ \hline N^\dagger & M \end{array} \right), \tag{17}$$

with $M$ and $N$ being $2n \times 2n$ matrices (and $M = M^\dagger$), it is always possible to define a density matrix $\rho^{x,k}$ such that Eq. (15) holds true.

This picture semiclassically describes the initial conditions. Now we consider the time evolution: to each ancillary fermion we associate a velocity through the correspondence (13)

$$f_i^{x,k} \to \mathbf{v}_i^k = v_i^k(k), \qquad f_{i+n}^{x,k} \to \mathbf{v}_{i+n}^k = v_i^k(-k). \tag{18}$$

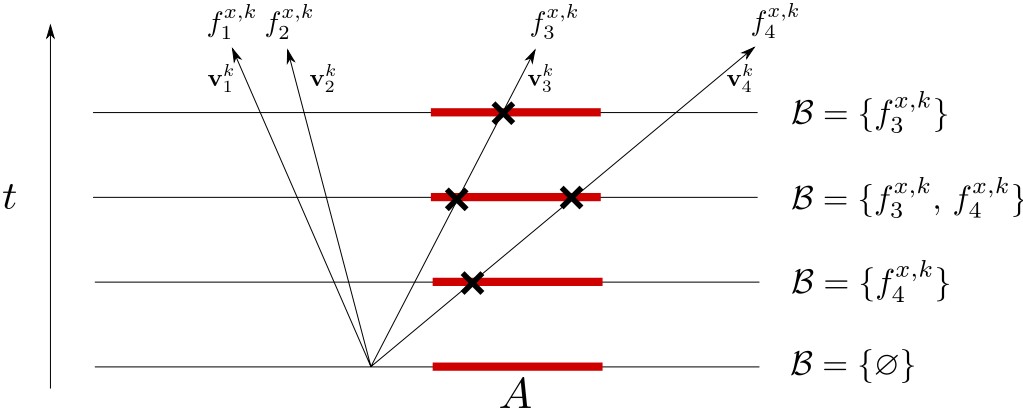

Figure 1: Quasiparticles which contribute to the entanglement entropy at time $t$. We consider a bipartition of the system $A \cup \bar{A}$ where $A$ is a single interval (red thick line). We show the ballistic evolution of some pairs of quasiparticles with momentum $k$ and originated in position $x$. Each fermion $f_i^{x,k}$ propagates with its own velocity $\mathbf{v}_i^k$. At a given time $t$, the initial set of fermions is divided into two subsets $\mathcal{B}$ and $\bar{\mathcal{B}}$, where in $\mathcal{B}$ appear those fermions which are carried in $A$ by the ballistic evolution.

Then, at a given time $t$, we introduce a bipartition of the ancillary Hilbert space $\mathcal{B} \cup \bar{\mathcal{B}}$ accordingly to the following rule (see also Fig. 1)

$$i \in \mathcal{B} \qquad \Longleftrightarrow \qquad x + t\mathbf{v}_i^k \in A. \tag{19}$$

We set $s_A(x, k, t)$ as the entanglement entropy of such a bipartition, i.e. we construct the reduced density matrix $\rho_{\mathcal{B}}^{x,k} = \text{Tr}_{\bar{\mathcal{B}}}[\rho^{x,k}]$ and pose

$$s_A(x, k, t) = -\text{Tr}_{\mathcal{B}}\big[\rho_{\mathcal{B}}^{x,k} \log(\rho_{\mathcal{B}}^{x,k})\big]. \tag{20}$$

Notice that, without explicitly computing $\rho_{\mathcal{B}}^{x,k}$, we can take advantage of the gaussianity of the reduced density matrix and express the Von Neumann entropy in terms of the correlation matrix [91–93]. In particular, let $\mathcal{C}^{\mathcal{B}}(k)$ be the correlation matrix extracted from $\mathcal{C}(k)$ in (16) retaining only those degrees of freedom in the $\mathcal{B}$ subspace, then it holds

$$s_A(x, k, t) = -\text{Tr}\big[\mathcal{C}^{\mathcal{B}}(k) \log \mathcal{C}^{\mathcal{B}}(k)\big]. \tag{21}$$

The equivalence between Eqs. (20) and (21) is discussed in Appendix A. Notice that the traces in Eqs. (20) and (21) are on very different spaces.

In analogy to Eq. (5), we can explicitly perform the integration over $x$ in the case when $A$ is an interval of length $\ell$. For simplicity, we assume that the quasiparticles are ordered in such a way that $\mathbf{v}_i^k > \mathbf{v}_j^k$ if $i < j$. This does not imply a loss of generality, since when it is not the case we can always, at fixed momentum, reorder the quasiparticles in such a way this requirement holds true, at the price that the needed reordering is momentum-dependent. Under these assumptions we have

$$S_A(t) = -\int_0^B \frac{\text{d}k}{2\pi} \sum_{i=1}^{2n} \sum_{j=1}^{i} \max\Big[0, \min[-t\mathbf{v}_{i+1}^k, \ell - t\mathbf{v}_j^k] - \max[-t\mathbf{v}_i^k, \ell - t\mathbf{v}_{j-1}^k]\Big]$$
$$\times \text{Tr}\big[\mathcal{C}^{\mathcal{B}_{j,i}} \log \mathcal{C}^{\mathcal{B}_{j,i}}\big], \tag{22}$$

where we conventionally set $\mathbf{v}_0^k = \infty$ and $\mathbf{v}_{2n+1}^k = -\infty$. The set of indexes $\mathcal{B}_{j,i}$ that must be extracted from the two-point correlation matrix is

$$\mathcal{B}_{j,i} = (j, j+1, ..., i) \cup (j+n, j+n+1, ..., i+n). \tag{23}$$

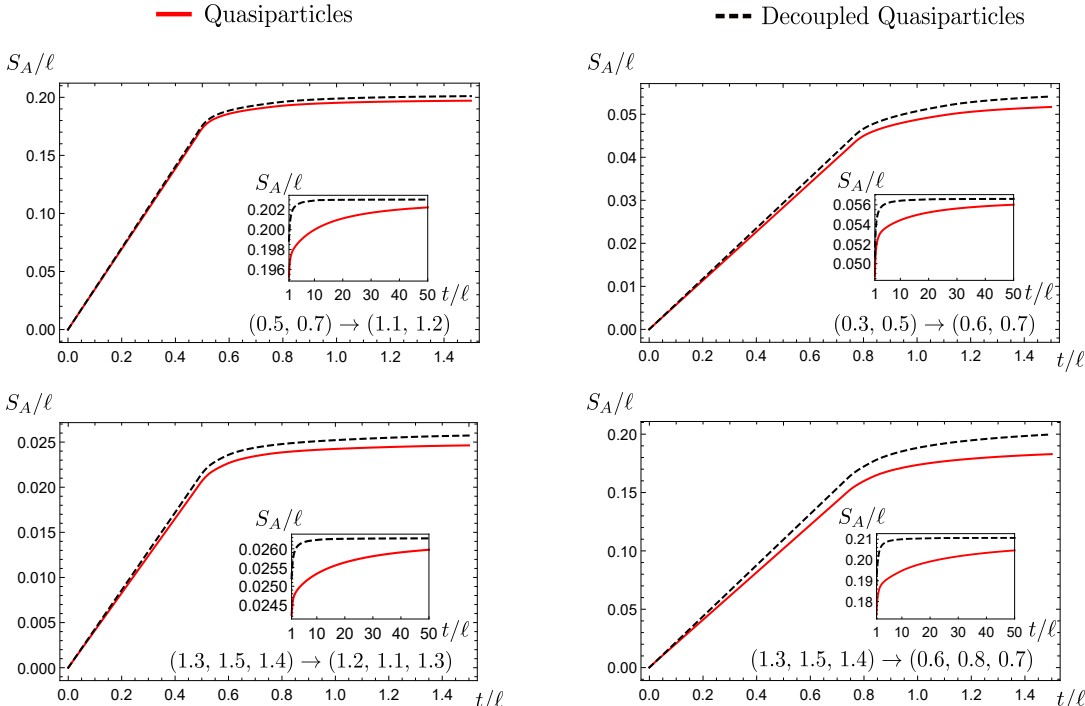

Figure 2: Entanglement entropy evolution for various quenches $\{h_i^{t<0}\}_{i=1}^n \rightarrow \{h_i\}_{i=1}^n$ in the inhomogeneous Ising model (further details in Section 3 ), for a bipartition $A \cup \bar{A}$, where $A$ is a finite interval of length $\ell$. In each panel we plot the rescaled entanglement entropy $S_A/\ell$ as a function of the rescaled time $t/\ell$ and compare our ansatz (continuous red line) with a naive application of the uncorrelated quasiparticle formula (5) (black dashed line), finding sizeable differences. At infinite time, the two predictions approach the same (thermodynamic) value as it should be (see insets).

The prediction to the entanglement growth provided by our ansatz quantitatively differs from the case where correlations are ignored (i.e., Eq. (5) extended to several species). This is clearly shown in Fig. 2, where we provide a few explicit examples anticipating our analysis of the Ising model of Section 3. Notice that although the two curves for correlated and uncorrelated pairs are quantitatively different, the light-cone spreading of entanglement still occurs even in the presence of correlations, as manifested by an initial linear increase followed by saturation to an extensive value in $\ell$ (which must be the same in the two cases). Yet, the growth rate of the entanglement for $2v_{\max}t < \ell$ is different.

A very important physical feature of the new prediction (22) is that it cannot be rewritten only in terms of the mode populations $n_i(k)$ and velocities $v_i(k)$, but the correlations in the initial state must be taken into account. Thus, contrarily to the standard uncorrelated case [60], the knowledge of the stationary state is not enough to fix the entire time dependence of the entanglement entropy.

## 2.1 Consistency checks

Given that our prediction (21) for the time evolution of the entanglement entropy is after all just a well thought conjecture, its validity must be ultimately tested against ab-initio calculations (either exact or numerical simulations). However, we can perform some non-trivial consistency checks comparing with the already well-established literature. First, we remark that, since the initial state (8) is pure, the density matrix associated with the ancillary Hilbert

space $\rho^{x,k}$ corresponds to a pure state as well, i.e. $\rho^{x,k} = |\Psi^{x,k}\rangle\langle\Psi^{x,k}|$, for a certain state $|\Psi^{x,k}\rangle$. This automatically guarantees the following properties.

1. The entanglement entropy $S_A(t)$ must be symmetric under exchange $A \leftrightarrow \bar{A}$. In Eq. (20) this follows from the facts that exchanging $A$ with $\bar{A}$ is equivalent to $\mathcal{B} \leftrightarrow \bar{\mathcal{B}}$ and that for any set of pairs with weight $s_A(x,k,t)$ Eq. (20) is symmetric under this operation.

2. Consider a given set of pairs and their weight $s_A(x,k,t)$: if all the particles at time $t$ belong to the same set (either $A$ or $\bar{A}$), we must have $s_A(x,k,t) = 0$. This is immediately guaranteed by the fact that $\text{Tr}\big[\rho^{x,k}\log\rho^{x,k}\big] = 0$, being $\rho^{x,k}$ associated with a pure state.

3. If only one particle species is present, then we must recover the standard quasiparticle prediction [38]. In the single species case, $\mathcal{C}^{(1)}(k)$ and $\mathcal{C}^{(2)}(k)$ are simple numbers, moreover $\mathcal{C}^{(1)}(k)$ is, by definition, the density of excitations $\mathcal{C}^{(1)}(k) = n(k)$. Thus the block matrix $\mathcal{C}(k)$ (16) specialised to a single species case reads

$$
\mathcal{C}(k) = \begin{pmatrix} 1-n(k) & 0 & 0 & [\mathcal{C}^{(2)}(-k)]^\dagger \\ 0 & 1-n(-k) & [\mathcal{C}^{(2)}(k)]^\dagger & 0 \\ 0 & \mathcal{C}^{(2)}(k) & n(k) & 0 \\ \mathcal{C}^{(2)}(-k) & 0 & 0 & n(-k) \end{pmatrix}.
\tag{24}
$$

When the quasiparticle of momentum $k$ is in $A$, while the companion at momentum $-k$ belongs to $\bar{A}$, the reduced correlation matrix is

$$
\mathcal{C}^{\mathcal{B}} = \begin{pmatrix} 1-n(k) & 0 \\ 0 & n(k) \end{pmatrix}.
\tag{25}
$$

Thus, from Eq. (21) we get

$$
s_A(x,k,t) = [-n(k)\log n(k) - (1-n(k))\log(1-n(k))]\Big|_{\substack{x+v_k t \in A \\ x+v_{-k} t \in \bar{A}}},
\tag{26}
$$

which coincides with the single species weight Eq. (6). Then, in the case where we are interested in a single interval, Eq. (5) is readily recovered from Eq. (22).

4. In the case of several particle species, but *uncorrelated* (i.e. $\mathcal{C}^{(1)}(k)$ and $\mathcal{C}^{(2)}(k)$ are diagonal), the generalisation of Eq. (5) to many species is readily obtained, as a straightforward extension of the single-species case.

5. At infinite time and choosing $A$ to be a finite interval, we must recover the Von Neumann entropy constructed on the late time steady state, i.e. it must hold true

$$
\lim_{t\to\infty} S_A(t) = \ell \sum_i \int_{-B}^{B} \frac{dk}{2\pi} s_i(k),
\tag{27}
$$

where

$$
s_i(k) = -n_i(k)\log n_i(k) - \big[1-n_i(k)\big]\log\big[1-n_i(k)\big].
\tag{28}
$$

This property is simply proven: for a very large time, looking at a set of pairs originated in $(x,k)$, at most one quasiparticle belongs to $A$ [94]. The distance between two quasiparticles associated with fermions $f_i^{x,k}$ and $f_j^{x,k}$ is $t|\mathbf{v}_i^k - \mathbf{v}_j^k|$, thus if $t > \ell/|\mathbf{v}_i^k - \mathbf{v}_j^k|$ they cannot both belong to $A$ (under the assumption of absence of velocity degeneracies). Therefore, if only the fermion $f_i^{x,k}$ belongs to $A$, constructing the reduced correlation matrix we find exactly Eq. (25) with $n(k) \to n_i(k)$. Thus, $s_A(x,k,t)$ reduces to Eq. (28). Considering the spatial integration in Eq. (12) we simply get a prefactor $\ell$ and Eq. (27) immediately follows.

## 3 The inhomogeneous Ising model

We now discuss the solution of the inhomogeneous Ising model (10) which provides a benchmark for our ansatz. We introduce fermionic degrees of freedom $\{d_j, d_{j'}^\dagger\} = \delta_{j,j'}$ through a Jordan Wigner transformation

$$d_j = e^{i\pi \sum_{l=1}^{j-1} \sigma_l^+ \sigma_l^-} \sigma_j^+, \qquad \sigma_j^\pm = (\sigma_j^x \pm i\sigma_j^y)/2. \tag{29}$$

In fermionic variables, the Ising Hamiltonian (10) can be rewritten as

$$H = \sum_{j=1}^N \left[ -\frac{1}{2}\left(d_j^\dagger d_{j+1}^\dagger + d_j^\dagger d_{j+1} + \text{h.c.}\right) + h_j d_j^\dagger d_j \right] + \text{boundary terms}. \tag{30}$$

Hereafter, we are interested in the thermodynamic limit $N \to \infty$ so that the boundary terms do not play any role and can be discarded. In order to diagonalise the Hamiltonian, it is convenient to move to Fourier space and define

$$d_j = \int_0^{2\pi} \frac{\mathrm{d}k}{\sqrt{2\pi}} e^{ikj} \alpha(k), \tag{31}$$

where $\{\alpha_k, \alpha_q^\dagger\} = \delta(k-q)$. In Fourier space, the Hamiltonian reads

$$H = -\int_0^{2\pi} \mathrm{d}k \frac{1}{2}\left(e^{ik}\alpha^\dagger(k)\alpha(k) + e^{ik}\alpha^\dagger(k)\alpha^\dagger(2\pi-k) + \text{h.c.}\right)$$
$$+ \int_0^{2\pi} \mathrm{d}k\mathrm{d}q\, \delta(e^{in(q-k)}-1)\tilde{h}(q-k)\alpha^\dagger(k)\alpha(q), \tag{32}$$

where we exploited the periodicity of $h_j$ and defined

$$\tilde{h}(k) = \sum_{j=0}^{n-1} h_j e^{ikj}. \tag{33}$$

In the Hamiltonian (32), the periodic field $h_j$ couples the modes accordingly to the "roots of unity rule". Now, we introduce several fermionic species splitting the Brillouin zone; in the momentum basis, we define $\beta_i(k)$ fermions as

$$\beta_i(k) = \alpha(k + (i-1)2\pi/n), \qquad i = 1\ldots n. \tag{34}$$

The momentum $k$ of the $\beta_i(k)$ fermions runs on the reduced Brillouin zone $k \in [0, 2\pi/n)$ and they satisfy standard anticommutation rules $\{\beta_i(k), \beta_j^\dagger(q)\} = \delta_{i,j}\delta(k-q)$. In terms of these fermions, the Hamiltonian becomes

$$H = \int_0^{2\pi/n} \mathrm{d}k \left[ -\frac{1}{2}\sum_{j=1}^n \left( e^{ik+(j-1)2\pi/n}\beta_j(k)^\dagger \beta_j(k) + e^{ik+(j-1)2\pi/n}\beta_j(k)^\dagger \beta_{n-j}^\dagger(2\pi/n-k) + \text{h.c.}\right) \right.$$
$$\left. + \sum_{i,j=1}^n \frac{1}{n}\tilde{h}((j-i)2\pi/n)\beta_i^\dagger(k)\beta_j(k) \right]. \tag{35}$$

We get a more compact notation organising the $\beta_i(k)$ fermions in an unique vector as

$$B^\dagger(k) = \left( \beta_1^\dagger(k), \beta_2^\dagger(k), ..., \beta_n^\dagger(k), \beta_1(2\pi/n-k), \beta_2(2\pi/n-k), ..., \beta_n(2\pi/n-k) \right), \tag{36}$$

and writing the Hamiltonian as

$$H = \int_0^{\pi/n} dk\, B^\dagger(k)\mathcal{H}^h(k)B(k).$$ (37)

The matrix $\mathcal{H}^h(k)$ can be written as

$$\mathcal{H}^h(k) = T(k) + \mathfrak{h}, \qquad T(k) = \left(\begin{array}{c|c} T_d(k) & T_{od}(k) \\ \hline T_{od}^\dagger(k) & -T_d(2\pi - k) \end{array}\right), \qquad \mathfrak{h} = \left(\begin{array}{c|c} \mathfrak{h}_d & 0 \\ \hline 0 & -\mathfrak{h}_d^* \end{array}\right),$$ (38)

where the matrix elements of the $n \times n$ blocks are

$$[T_d(q)]_{a,b} = -\delta_{a,b}\cos\left(q + 2\pi\frac{a}{n}\right),$$

$$[T_{od}(q)]_{a,b} = -i\delta_{n-1-a,b}\sin\left(q + 2\pi\frac{a}{n}\right),$$

$$[\mathfrak{h}_d]_{a,b} = \frac{1}{n}\tilde{h}\left(\frac{2\pi}{n}(b-a)\right).$$ (39)

The desired modes are identified through the diagonalisation of $\mathcal{H}^h(k)$. Indeed, given the block form of $\mathcal{H}^h(k)$, it always exists an unitary transformation $U^h(k)$ such that

$$[U^h(k)]^\dagger \mathcal{H}^h(k)U^h(k) = \left(\begin{array}{c|c} \mathcal{E}^h(k) & 0 \\ \hline 0 & -\mathcal{E}^h(k) \end{array}\right).$$ (40)

Here $\mathcal{E}^h(k)$ are positive defined diagonal matrices, which are the energies of the modes

$$[\mathcal{E}^h(k)]_{i,j} = \delta_{i,j}E_i(k).$$ (41)

The modes $\eta_i(k)$ are then identified as the solution of the linear equation

$$B(k) = U^h(k)G^h(k),$$ (42)

where $G^h(k)$ is defined as

$$\left[G^h(k)\right]^\dagger = \left(\eta_1^\dagger(k), \eta_2^\dagger(k), ..., \eta_n^\dagger(k), \eta_1(-k), \eta_2(-k), ..., \eta_n(-k)\right).$$ (43)

With this definition, we made the Brillouin zone symmetric around zero.

Having diagonalised the Hamiltonian for arbitrary magnetic field, we can now consider a quench changing the magnetic field from $\{h_j^{t<0}\}_{j=1}^n$ to $\{h_j\}_{j=1}^n$. Then the pre and post quench modes are connected through a proper Bogoliubov transformation. In particular, from Eq. (42) we have

$$U^{h^{t<0}}(k)G^{h^{t<0}}(k) = B(k) = U^h(k)G^h(k) \implies G^{h^{t<0}}(k) = \left[\left(U^{h^{t<0}}(k)\right)^\dagger U^h(k)\right]G^h(k).$$ (44)

We finally need to show that the initial state (i.e. the ground state $\{h_j^{t<0}\}_{j=1}^n$), when expressed in the post quench modes for a magnetic field $\{h_j\}_{j=1}^n$, is of the form in Eq. (8). The initial state is the vacuum for the prequench modes, therefore, using Eq. (44), we can readily write the set of equations

$$\left(\sum_{j=1}^n \mathcal{U}_{i,j}(k)\eta_j(k) + \sum_{j=1}^n \mathcal{U}_{i,j+n}(k)\eta_j^\dagger(-k)\right)|0_{h^{t<0}}\rangle = 0, \qquad \forall i \in \{1,...,n\},$$ (45)



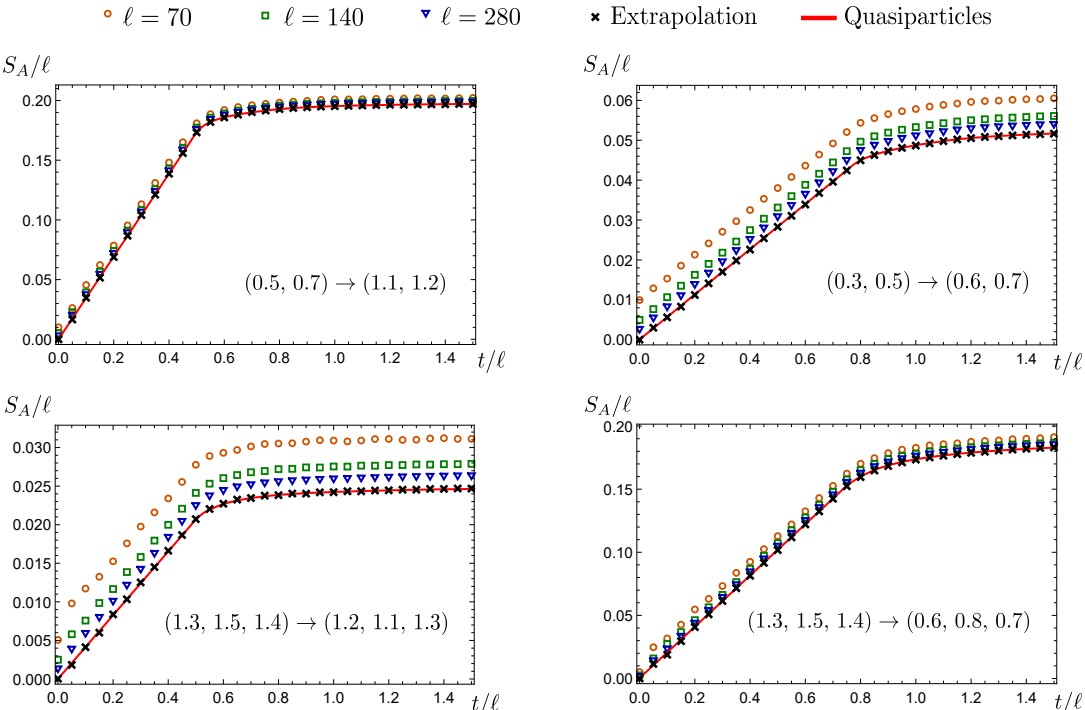

Figure 3: Entanglement entropy evolution for various quenches $\{h_i^{t<0}\}_{i=1}^n \rightarrow \{h_i\}_{i=1}^n$ in the inhomogeneous Ising model. We consider a bipartition $A \cup \bar{A}$ where $A = [1, \ell]$. In each panel we plot the rescaled entanglement entropy $S_A/\ell$ as a function of the rescaled time $t/\ell$. When the finite $\ell$ data are extrapolated to $\ell \rightarrow \infty$, as explained in the text, the agreement with the quasiparticle prediction is perfect.

where for compactness we set

$$\mathcal{U}_{i,j}(k) = \left[ \left( U^{h^{t<0}}(k) \right)^\dagger U^h(k) \right]_{i,j}. \tag{46}$$

It is then straightforward to realise that $|0^{h^{t<0}}\rangle$ can be written in the form Eq. (8), i.e.

$$|0_{h^{t<0}}\rangle \propto \exp\left[ -\int_0^{\pi/n} dk \sum_{i=1,j=1}^n \mathcal{M}_{i,j}(k)\, \eta_i^\dagger(k)\eta_j^\dagger(-k) \right] |0_h\rangle, \tag{47}$$

provided the matrix $\mathcal{M}_{i,j}(k)$ satisfies the equation

$$\sum_{j=1}^n \mathcal{U}_{i,j}(k)\mathcal{M}_{j,i'}(k) = \mathcal{U}_{i,i'+n}(k), \qquad \forall i \in \{1, ..., n\}. \tag{48}$$

We can then conclude that quenches in the inhomogeneous Ising spin chain (10) fall within the framework of our ansatz for the entanglement spreading, which can now be tested. As we saw, finding the energies of the modes $E_i(k)$ and the $t = 0$ correlations of the post quench modes boils down to diagonalising finite-dimensional matrices; even though pushing further the analytical calculations can be cumbersome (especially if several species are involved), this last step can be quickly carried out numerically.

In Fig. 3 we test the ansatz against direct exact numerical calculations in the Ising model for various choices of the pre and post quench magnetic fields. Numerical calculations have been carried out on a lattice of 1200 sites with periodic boundary conditions, by mean of a

direct solution of the free fermion model. We consider time $t$ and subsystem sizes $\ell$ such that the finite size of the entire system does not play a role. We focus on a bipartition where $A$ is a finite interval of length $\ell$. The figure shows that as $\ell$ becomes larger, the numerical results clearly approach the quasiparticles ansatz. In order to provide a stronger evidence for the correctness of our conjecture we provide an extrapolation to $\ell \to \infty$. In this perspective, we assume a regular expansion in powers of $\ell^{-1}$

$$\frac{S_A(t/\ell)}{\ell} = s_A^{\text{QP}}(t/\ell) + \frac{1}{\ell} s_A^1(\ell, t) + \dots \tag{49}$$

where $s_A^{\text{QP}}$ is the quasiparticle prediction. Performing a fit of our data with the above form at order $\mathcal{O}(\ell^{-1})$, we obtain the extrapolation that are represented as crosses in the figure. It is evident that these extrapolations perfectly match the quasiparticle ansatz for all the considered quenches.

## 4  Time evolution of the order parameter correlations

Although the main focus of this work is the entanglement entropy, the quasiparticle picture may provide useful information even for other quantities, primarily the order parameter $\langle \sigma_j^x \rangle$ and its two-point correlation [63, 64]. For the quantum quench in the homogeneous Ising model, it has been shown that the quasiparticle prediction is qualitatively and quantitatively correct only for quenches within the ordered phase [30,31,44]. For quenches from the ordered phase to the paramagnetic one, the quasiparticle picture can be heuristically adapted to provide correct results [30,31]. Instead in the case of quenches starting from the paramagnetic phase, it is still not known whether it is possible to use these ideas to have an exact ansatz; anyhow a discussion of this issue is beyond the scope of this paper, see Ref. [30, 31]. Consequently, in this section we limit ourselves to extend the quasiparticle picture for the order parameter correlations to initial states with correlated quasiparticles for quenches *within the ferromagnetic phase* and to test the prediction in the inhomogeneous Ising chain (10).

Correlation functions of the order parameter are non local objects when expressed in the fermionic basis

$$\langle \sigma_j^x \sigma_{j'}^x \rangle = \left\langle (d_j^\dagger + d_j) e^{-i\pi \sum_{l=j+1}^{j'-1} d_l^\dagger d_l} (d_{j'}^\dagger + d_{j'}) \right\rangle. \tag{50}$$

Although the computation of the correlator (and its time evolution) ultimately boils down to an extensive use of the Wick theorem and evaluating determinants, the large number of the involved degrees of freedom makes the calculation very complicated. In Ref. [30, 31] a first-principle calculation was carried out in the homogeneous Ising model, resulting in a scaling behaviour that can be interpreted a posteriori in terms of quasiparticles. While in principle possible, we do not try to generalise the complicated methods of Ref. [30, 31], but, inspired by the resulting expression, we directly attempt a quasiparticle ansatz which is then numerically verified. A posteriori, we will see that our ansatz fails to describe those situations where $\langle \sigma_j^x \rangle = 0$ on the initial state, as expected on the basis of the results for the homogeneous case [30, 31].

Inspired by Ref. [30, 31], we conjecture the following ansatz for the logarithm of the two-point correlator

$$\log |\langle \sigma_j^x \sigma_{j'}^x \rangle| = \int_{-\infty}^{\infty} dx \int_0^B \frac{dk}{2\pi} \log(p_A(x, k, t)) + \dots \tag{51}$$

where $A$ is the interval of extrema $j$ and $j'$, the quantity $p_A(x, k, t)$ is discussed hereafter. An explicit space integration may eventually lead to a formula similar to the entropy one in Eq. (22).

Our ansatz, relays on the following assumptions, similar to those used in the entanglement entropy case: *i)* quasiparticles generated at different spatial points or belonging to pairs of different momenta are uncorrelated; *ii)* the large distance behaviour of the correlator Eq. (50) is ultimately determined by the string

$$e^{-i\pi \sum_{l=j+1}^{j'} d_l^\dagger d_l} = \prod_{l=j+1}^{j'} (1 - 2d_l^\dagger d_l). \tag{52}$$

(In passing: most likely *ii)* is the hypothesis that is failing for quenches from the disordered phase.) Consider then a semiclassical computation of $\langle e^{-i\pi \sum_{l=j+1}^{j'} d_l^\dagger d_l}\rangle$: since pairs originated at different position or having different momentum are uncorrelated, the expectation value should factorise in the contribution of each set of pairs. Equivalently, the logarithm must be additive, justifying the form of Eq. (51). Now, in order to find the proper ansatz for $p_A(x, k, t)$, we recognise that the string (52) simply counts the parity of the number of fermions within the interval $(j, j')$. Therefore, using the same notation of Section 2 for the auxiliary fermions $f_j^{x,k}$, we take as an ansatz

$$p_A(x, k, t) = \left| \left\langle \prod_{i=1}^{2n} \left[ 1 - 2(f_i^{x,k})^\dagger f_i^{x,k} \right]_{x + \mathbf{v}_i^k t \in A} \right\rangle \right|, \tag{53}$$

where in the product only those fermions which semiclassically belong to the interval $x + \mathbf{v}_i^k t \in A$ must be considered. The expectation value is taken on the auxiliary Hilbert space as in Section 2.

The time evolution of the order parameter itself may be accessed from Eq. (51) using the cluster decomposition principle, obtaining

$$\langle \sigma_j^x \sigma_{j'}^x \rangle \simeq \langle \sigma_j^x \rangle \langle \sigma_{j'}^x \rangle, \qquad |j - j'| \gg 1. \tag{54}$$

Under the further assumption that $\langle \sigma_j^x \rangle$ is translational invariant in the scaling limit (as it is the case here), we therefore obtain a quasiparticle prediction for the one point function. This observation, besides providing an additional result, also helps a posteriori to understand the regime of applicability of the quasiparticle ansatz. Indeed, from Eq. (53) $|p_A(x, k, t)| \le 1$ (it is a product of terms that are all smaller than 1). Therefore, the r.h.s. of Eq. (51) is surely negative (or at most zero), implying an exponentially decaying $|\langle \sigma_j^x \rangle|$, which cannot be correct if the order parameter is zero in the initial state.

We close this section mentioning that by the repeated use of the Wick Theorem, $p_A(x, k, t)$ can be efficiently formulated in terms of a determinant of a correlation function of the fermions. Assume that the fermions $f_{i_j}^{x,k}$ for a set of indexes $i_j$, $j \in \{1, ..., n'\}$ are those belonging to $A$. Then, we consider a $2n' \times 2n'$ antisymmetric matrix $\mathcal{A}$ defined as

$$\mathcal{A}_{2(j-1)+1, 2(j'-1)+1} = \begin{cases} \left\langle \left( f_{i_j}^{x,k} - (f_{i_j}^{x,k})^\dagger \right) \left( f_{i_{j'}}^{x,k} - (f_{i_{j'}}^{x,k})^\dagger \right) \right\rangle & j \neq j' \\ 0 & j = j' \end{cases} \tag{55}$$

$$\mathcal{A}_{2j, 2j'} = \begin{cases} \left\langle \left( f_{i_j}^{x,k} + (f_{i_j}^{x,k})^\dagger \right) \left( f_{i_{j'}}^{x,k} + (f_{i_{j'}}^{x,k})^\dagger \right) \right\rangle & j \neq j' \\ 0 & j = j' \end{cases} \tag{56}$$

$$\mathcal{A}_{2(j-1)+1, 2j'} = -\mathcal{A}_{2j', 2(j-1)+1} = \left\langle \left( f_{i_j}^{x,k} - (f_{i_j}^{x,k})^\dagger \right) \left( f_{i_{j'}}^{x,k} + (f_{i_{j'}}^{x,k})^\dagger \right) \right\rangle. \tag{57}$$

In terms of the matrix $\mathcal{A}$, it holds (see Appendix B)

$$p_A(x, k, t) = \sqrt{\det \mathcal{A}}. \tag{58}$$

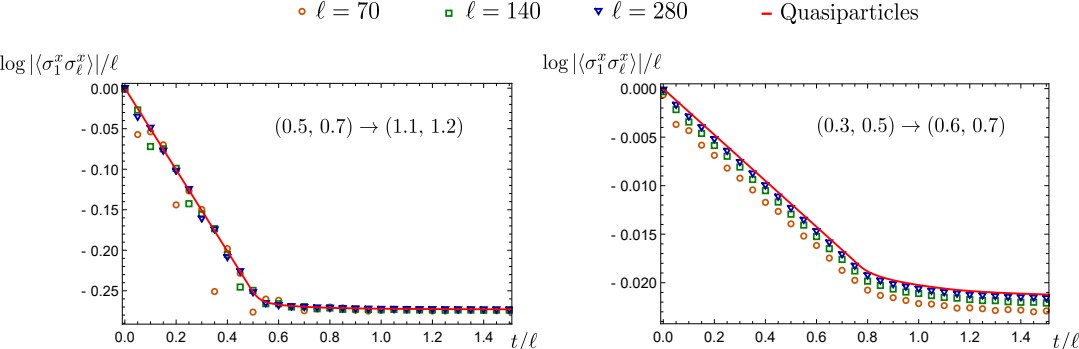

Figure 4: Evolution of the two-point correlator for various quenches $\{h_i^{t<0}\}_{i=1}^n \to \{h_i\}_{i=1}^n$ in the inhomogeneous Ising model. We test the time evolution of the two-point correlator of the order parameter against the quasiparticle ansatz for increasing separation $\ell$. It is evident that increasing $\ell$ the numerical data quickly approach the quasiparticle ansatz. The initial exponential decay is due to the evolution of the one-point function of the order parameter since, on that time scale, we have $\langle \sigma_1^x \sigma_\ell^x \rangle \simeq \langle \sigma_1^x \rangle \langle \sigma_\ell^x \rangle$. At late times, saturation to the steady state value is observed.

We tested our ansatz against exact numerical calculations for quenches in the inhomogeneous Ising chain. We found that for all quenches within the ferromagnetic phase, our quasiparticle prediction perfectly reproduces the numerical data in the space-time scaling limit $t, \ell \to \infty$ with $t/\ell$ fixed. Two representative cases are shown in Fig. 4. It is evident that by increasing $\ell$ the numerical data approach the prediction, although finite size correction are clearly visible at small $\ell$, but these are much smaller than those for the entanglement entropy. It is clear from the result that even for the two-point function of the order parameter, the "light-cone" spreading of correlations persists in the presence of correlated quasi-particles. We also checked that for other quenches (i.e. from and to the paramagnetic phase) the conjecture does not work, as expected. Finally, to be exhaustive, we remind the reader that the quasiparticle prediction is correct for quenches to the critical point from the ordered phase, but not for quenches originating from the critical point [30, 31].

## 5 Conclusions

In this manuscript we generalised the semiclassical quasiparticle picture for the spreading of entanglement and correlations to global quantum quenches with multiple species of quasiparticles that show momentum-pair correlations in the initial state. The main new physical result (compared to the standard uncorrelated case) is that the information encoded in the mode populations $n_i(k)$ of the single species and their velocities $v_i(k)$ are not enough to determine the time evolution of the entanglement entropy and correlations. We show that among the systems displaying this phenomenology for the quasiparticles, a remarkable example is the Ising chain with a periodically-modulated transverse field, which is easily mappable to a free fermionic theory. We then use this model to test our predictions for entanglement and correlations against exact numerical calculations, finding perfect agreement in the scaling regime.

A simple and straightforward generalisation of our results concerns the case when the initial state is inhomogeneous on a large scale so to be describable by the generalised hydrodynamics approach [95–103]. For example, we have in mind the joining of two different thermal states or groundstates of different Hamiltonians producing correlated quasiparticles. In this case, it is very simple to merge the results of the present paper with those of Ref. [51].

The spatial variation of the entanglement entropy may be properly captured by a term that locally is given by Eq. (21).

Another inhomogeneous setup in which several correlated pairs can be produced is that of moving defects [104, 105], where an external localised perturbation is dragged at constant velocity in an otherwise homogeneous free lattice system. The moving impurity can be regarded as a source of quasiparticles and the propagation of entanglement could fall within our frame, again supplemented with the generalised hydrodynamics [95–103].

A simple generalisation of our results is the time evolution of Rényi entanglement entropies that for free models just requires a minor variation of the form of the kernel (21), see for example the discussion in [106].

Finally, a more difficult open problem concerns the spreading of entanglement in interacting integrable models. For these models there are almost always multiple species of quasiparticles (they are bound states of the lightest species), but known integrable initial states do not have correlations between them and the general evolution of the entanglement entropy has been understood in [60–62] (for the Rényi entropies see [106–109] while for inhomogeneous systems see [110]). This lack of correlations in solvable initial states has been sometimes related to the integrability of the quench problem itself [111, 112]. Yet, it would be very interesting to understand whether, at least in in some models, it is possible to have correlated quasiparticles which would require the generalisation of our approach to interacting integrable systems.

### Acknowledgments

P. C. acknowledges support from ERC under Consolidator grant number 771536 (NEMO). Part of this work has been carried out during the workshop "Quantum Paths" at the Erwin Schrödinger International Institute for Mathematics and Physics (ESI) in Vienna, and during the workshop "Entanglement in Quantum Systems" at the Galileo Galilei Institute (GGI) in Florence.

## A  Von Neumann entropies in gaussian states

The rewriting of the Von Neumann entanglement entropy in terms of the correlation matrix of a Gaussian state is a well understood subject [91–93] that we briefly review in the following also to fix our convention. We consider $n'$ fermions $f_i$ with $i \in \{1, ..., n'\}$ satisfying canonical anticommutation relations $\{f_i, f_j\} = \delta_{i,j}$ (here $n'$ incorporates all kinds of fermion indexes, e.g. species, lattice sites, etc). We consider a Gaussian state (i.e. the Wick Theorem holds) described by a density matrix $\rho$ with two-body correlation matrix is $C \equiv \langle F F^\dagger \rangle$, where as usual we have defined

$$F^\dagger = (f_1^\dagger, \ldots, f_{n'}^\dagger, f_1, \ldots, f_{n'}). \tag{59}$$

The Von Neumann entropy associated with $\rho$ can be written as

$$-\mathrm{Tr}[\rho \log \rho] = -\mathrm{Tr}\Big[C \log C\Big], \tag{60}$$

as we are going to show. The main advantage of Eq. (60) is that while the trace on the left hand side is taken on the $2^{n'}$-dimensional Hilbert space, the trace on the right is instead on the indexes of the matrix $C$, i.e. just on a $2n'$-dimensional space. The equality is readily proven diagonalising $C$: it exists an unitary matrix $U$ associated with a Bogoliubov rotation such that

$$\mathcal{U}^\dagger C \mathcal{U} = \left( \begin{array}{c|c} \mathrm{Id} - \mathcal{D} & 0 \\ \hline 0 & \mathcal{D} \end{array} \right), \tag{61}$$

with $\mathcal{D}_{i,j} = \delta_{i,j}\Delta_j$ a diagonal matrix. Using $U$ we can define new fermionic operators $\tilde{f}_i$ such that the correlator is diagonal

$$F = U\tilde{F}, \qquad \langle \tilde{f}_i^\dagger \tilde{f}_j \rangle = \delta_{i,j}\Delta_j. \tag{62}$$

The gaussianity of the ensemble and the diagonal correlator, necessarily implies that the density matrix in the new basis is written as

$$\rho \propto e^{-\sum_{i=1}^{n'} \epsilon_i \tilde{f}_i^\dagger \tilde{f}_i}, \qquad \Delta_j = \frac{1}{e^{\epsilon_j} + 1}. \tag{63}$$

Hence the entanglement entropy easily follows

$$-\text{Tr}[\rho \log \rho] = \sum_{j=1}^{n'} \Big[ -\Delta_j \log \Delta_j - (1-\Delta_j)\log(1-\Delta_j) \Big]. \tag{64}$$

To recover Eq. (60) it is enough to notice that the spectrum of $C$ consists in $\{\Delta_i\}_{i=1}^{n'} \cup \{1-\Delta_i\}_{i=1}^{n'}$. Therefore we can equivalently write

$$-\text{Tr}[\rho \log \rho] = \sum_{\lambda_l \text{ eigenvalue of } C} \Big[ -\lambda_l \log \lambda_l \Big], \tag{65}$$

which can be written in a basis-independent way as per Eq. (60).

# B  Proof of Eq. (58)

Eq. (58) can be proven, e.g., following Ref. [30, 31]. Hereafter, we consider the general problem of determining

$$p = \left| \left\langle \prod_{i=1}^{n'} \left(1 - 2f_i^\dagger f_i\right) \right\rangle \right|, \tag{66}$$

where $f_i$ are fermionic operators $\{f_i, f_j^\dagger\} = \delta_{i,j}$ and the state is Gaussian in these fields. We first rewrite $p$ as

$$p = \left| \left\langle \prod_{i=1}^{n'} \left(f_i - f_i^\dagger\right)\left(f_i + f_i^\dagger\right) \right\rangle \right| = \left| \left\langle \prod_{i=1}^{2n'} b_i \right\rangle \right|, \tag{67}$$

where we naturally introduced the operators $b_i$ as

$$b_{2(i-1)+1} = f_i - f_i^\dagger, \qquad b_{2i} = f_i + f_i^\dagger. \tag{68}$$

It holds true $\{b_i, b_j\} = 0$ for $i \neq j$. Let us introduce the following antisymmetric matrix $A_{i,j}$

$$A_{i,j} = \begin{cases} \langle b_i b_j \rangle & i \neq j \\ 0 & i = j \end{cases}. \tag{69}$$

Therefore, as noticed in Ref. [30,31], by mean of a simple application of the Wick Theorem it can be shown

$$\left\langle \prod_{i=1}^{2n'} b_i \right\rangle = \text{Pf}(A), \tag{70}$$

where $\text{Pf}(A)$ is the Pfaffian of the matrix $A$. Given that $|\text{Pf}(A)| = \sqrt{\det(A)}$, this concludes the proof. Indeed, the definition of the matrix $\mathcal{A}$ in Eqs. (55,56,57) of Section 4 is nothing else that the analogue of the matrix $A$.

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
