# Peer review of "Spreading of entanglement and correlations after a quench with intertwined quasiparticles"

_SciPost Physics, doi:SciPost Phys. 5, 033 (2018)_

## Round 1 · Referee Report · Anonymous (Referee 1) · 2018-9-5

Strengths

The paper describes the spreading of entanglement entropy following a quantum quench in the case in which the model Hamiltonian admits several quasi-particle excitations.

1) The paper is clearly written. 2) The semiclassical ansatz is well presented and supported by additional explicit calculations for a Ising-like chain. 3) The message is sharp and easy to understand. 4) The bibliography well prepared.

Weaknesses

no specific points

Report

In my opinion the paper contains interesting results. In some aspects this is a generalisation to the case of several types of excitations. It should be said that this generalisation contains some non-trivial aspects that are worth being published.

Requested changes

no changes

  • validity: top
  • significance: high
  • originality: good
  • clarity: top
  • formatting: excellent
  • grammar: excellent

Author:  Alvise Bastianello  on 2018-09-26  [id 322]

(in reply to Report 1 on 2018-09-05)

We are grateful to Referee 1 for his/her appreciation of our work and the opinion it is worth to be published.

---

## Round 1 · Referee Report · Anonymous (Referee 2) · 2018-9-24

Strengths

The paper is well written, topical, and has a clear message.

Weaknesses

None.

Report

In the manuscript "Spreading of entanglement and correlations after a quench with intertwined quasiparticles" the authors consider the problem of describing in a semiclassical fashion the non-equilibrium post-quench dynamics of a system with multiple species of particles. The models they consider are quadratic, but where the pre-quench state of the system possess non-trivial pair correlations in momentum space. In describing the system semi-classically, the authors wish to show the a picture based on quasi-particle generation at the time of the quench and subsequent propagation is able to describe the growth of entanglement entropy and the evolution of the order parameter. As a test of their formalism, they consider quenches involving a quantum Ising model with a periodically varying transverse magnetic field in its ordered phase.

I think this paper is a useful contribution to our understanding of when and how a semiclassical picture can be used in the description of non-equilibrium dynamics of quantum systems. I particularly like the non-trivial extension of the quasi-particle picture to a case where correlations are present. I think this paper should be accepted for publication.

I do have one substantive comment that the authors may want to address:

In the treatment of quenches with an initial state that has non-trivial correlations, they still suppose that that effective set of quasi-particles does not see correlations between different momenta. Of course in fact in the example that they consider there are correlations between particles of different momenta because of the periodic nature of the transverse field and this inter-momenta correlations is subsequently "hidden" by partitioning the original Brillouin zone into n parts
(where n is the periodicity of the transverse field) and introducing a set of quasi-particles in this fragmented k-space.

My question is then do the authors think that finding a basis of effective quasi-particles where the correlations are only between different species of quasi-particles, but with same momentum, is a necessary condition for the quasi-particle picture to be valid. Could you, for example, initialize the Ising model in the ground state belonging to a random transverse field and then do the post-quench evolution in the periodic transverse field model considered in the paper and expect some formulation of the quasi-particle to work? Such an initial state would presumably involve correlations between different quasi-particle momenta. I am not looking for a definitive answer here (this would likely involve a whole new paper), but it would be interesting if the authors could comment on this possibility in the text of the current manuscript.

Requested changes

I spotted a number of minor typos:

Above Eqn. 2: "dragging" -> "deriving"

Below Eqn. 4: "originated" x 2 -> "originating" x 2

Below Eqn. 4: "considered" -> "considering"

Below Eqn. 4: "being their contribution" -> "their contribution being"

Below Eqn. 9: "can reside to" -> "can find"

Above Eqn. 53: "pose" -> "take as an ansatz"

page 13: "is drag at" -> "is dragged at"

  • validity: top
  • significance: high
  • originality: high
  • clarity: top
  • formatting: excellent
  • grammar: good

Author:  Alvise Bastianello  on 2018-09-26  [id 323]

(in reply to Report 2 on 2018-09-24)

We thank Referee 2 for his/her appreciation of our work, we are also grateful for having signaled us a few typos that we correct in the resubmitted version.

The referee asks about the possibility of extending the method to even more general quenches, asking for the specific example of a quench from locally random values of the magnetic field in the Ising chain.

The problem of determining the entanglement growth and the underlying quasiparticle description for general cases is of utmost importance and, as the referee commented, it requires further work and we do not have a satisfactory answer yet. For this reason, in the present manuscript we prefer to set aside vague comments and outlooks.

As a general comment, we expect the quasiparticle ansatz to have a wide range of applicability, but in practice the excitations' structure of the initial state must be known and simple enough. For example, in homogeneous free quenches the single pair structure is the common situations and, in the truly interacting integrable case, there are strong indications [J. Phys. A: Math. Theor. 47, 402001 (2014), J. Phys. A: Math. Theor. 50, 84004, Nucl. Phys. B 925 (2017) 362-402] that the only initial states which can be exactly determined are those with a pair structure (with uncorrelated species).
The construction of workable examples which lays beyond this paradigm required a certain degree of inventiveness, as in [J. Stat. Mech. (2018) 063104] and in the present work. Therefore, we leave further investigations for the future.

Coming back to the actual example asked by the referee, i.e. quenching the Ising chain with a random magnetic field, a first step for the understanding of the problem could be to consider the periodic case, but with larger and larger periods. A first drawback is the growing dimension of the auxiliary Hilbert space, with an increasing complexity in computing the entanglement weights needed for our ansatz. Furthermore, our quasiparticle ansatz holds true in the scaling region, i.e. for distances much larger than the period of the magnetic field.

---

## Editorial Decision

published